# Identification of Binding Proteins for TSC22D1 Family Proteins Using Mass Spectrometry

**DOI:** 10.3390/ijms222010913

**Published:** 2021-10-09

**Authors:** Ryouta Kamimura, Daisuke Uchida, Shin-ichiro Kanno, Ryo Shiraishi, Toshiki Hyodo, Yuta Sawatani, Michiko Shimura, Tomonori Hasegawa, Maki Tsubura-Okubo, Erika Yaguchi, Yuske Komiyama, Chonji Fukumoto, Sayaka Izumi, Atsushi Fujita, Takahiro Wakui, Hitoshi Kawamata

**Affiliations:** 1Department of Oral and Maxillofacial Surgery, Dokkyo Medical University School of Medicine, 880 Kita-kobayashi, Shimotsuga, Mibu 321-0293, Tochigi, Japan; kmry28@dokkyomed.ac.jp (R.K.); ryo-s@dokkyomed.ac.jp (R.S.); hyodo14@dokkyomed.ac.jp (T.H.); sawayu@dokkyomed.ac.jp (Y.S.); smichiko@dokkyomed.ac.jp (M.S.); hase-t@dokkyomed.ac.jp (T.H.); makio@dokkyomed.ac.jp (M.T.-O.); eri-yagu@dokkyomed.ac.jp (E.Y.); y-komi@dokkyomed.ac.jp (Y.K.); chonji-f@dokkyomed.ac.jp (C.F.); saya@dokkyomed.ac.jp (S.I.); fujita-a@dokkyomed.ac.jp (A.F.); t-wakui@dokkyomed.ac.jp (T.W.); 2Department of Oral and Maxillofacial Surgery, Ehime University Graduate School of Medicine, Shitsukawa, Toon 791-0295, Ehime, Japan; udai@m.ehime-u.ac.jp; 3Division of Dynamic Proteome, Institute of Development, Aging, and Cancer, Tohoku University, 4-1 Seiryo-machi, Sendai 980-8575, Aobaku, Japan; shinichiro.kanno.a6@tohoku.ac.jp; 4Section of Dentistry, Oral and Maxillofacial Surgery, Kamitsuga General Hospital, 1-1033 Shimoda-machi, Kanuma 322-8550, Tochigi, Japan; 5Section of Dentistry and Oral and Maxillofacial Surgery, Sano Kosei General Hospital, 1728 Horigomecho, Sano 327-8511, Tochigi, Japan

**Keywords:** TSC22D1 family proteins, binding protein, mass spectrometry, Histone H1, Guanine nucleotide-binding protein-like 3 (GNL3), molecular-targeted therapy, differentiation-inducing therapy

## Abstract

TSC-22 (TGF-β stimulated clone-22) has been reported to induce differentiation, growth inhibition, and apoptosis in various cells. TSC-22 is a member of a family in which many proteins are produced from four different family genes. TSC-22 (corresponding to TSC22D1-2) is composed of 144 amino acids translated from a short variant mRNA of the TSC22D1 gene. In this study, we attempted to determine the intracellular localizations of the TSC22D1 family proteins (TSC22D1-1, TSC-22 (TSC22D1-2), and TSC22(86) (TSC22D1-3)) and identify the binding proteins for TSC22D1 family proteins by mass spectrometry. We determined that TSC22D1-1 was mostly localized in the nucleus, TSC-22 (TSC22D1-2) was localized in the cytoplasm, mainly in the mitochondria and translocated from the cytoplasm to the nucleus after DNA damage, and TSC22(86) (TSC22D1-3) was localized in both the cytoplasm and nucleus. We identified multiple candidates of binding proteins for TSC22D1 family proteins in in vitro pull-down assays and in vivo binding assays. Histone H1 bound to TSC-22 (TSC22D1-2) or TSC22(86) (TSC22D1-3) in the nucleus. Guanine nucleotide-binding protein-like 3 (GNL3), which is also known as nucleostemin, bound to TSC-22 (TSC22D1-2) in the nucleus. Further investigation of the interaction of the candidate binding proteins with TSC22D1 family proteins would clarify the biological roles of TSC22D1 family proteins in several cell systems.

## 1. Introduction

Many cancers including head and neck cancer have been treated by molecular-targeted drugs in addition to surgery, chemotherapy, and radiotherapy [1,2]. Acute promyelocytic leukemia can be treated by all-trans retinoic acid (ATRA), which is a molecular-targeted drug and has a differentiation-inducing activity [3,4]. This therapy results in the death or reduced growth of neoplastic cells by inducing differentiation and maturation through a course similar to that of normal cells, in contrast to classical anticancer drugs that are cytotoxic to neoplastic cells [5,6,7]. Clinical application of differentiation-inducing therapy has advanced in hematopoietic malignancies, but has not been established for solid tumors. This is partly because the mechanism and molecular regulation of cell differentiation is unclear in solid tumors.

In 1994, our group cloned TGFβ-stimulated clone-22 (TSC-22) cDNA as a gene induced by a differentiation-inducing drug in cultured salivary gland cancer cells (TYS cells), and we analyzed the TSC-22 genomic structure, regulation of the gene expression, and protein structure and function [8,9,10,11,12,13,14]. TSC-22 was shown to induce differentiation and inhibit proliferation in several cell types and to increase the susceptibility of TYS cells to chemotherapy and radiotherapy [10,12]. Furthermore, TSC-22 transgenic mice became severely obese and frequently developed B cell lymphoma [14]. In cells overexpressing TSC-22, the protein was localized in the cytoplasm by its nuclear export signal (NES) in the static state and was translocated into the nucleus in a stress state, such as cytotoxicity [13].

TSC-22 is part of a family in which many mRNAs with different lengths are transcribed from four different genes as splice variants, and many proteins are translated from these mRNAs [15,16]. The proteins reported to date are shown in Table 1 [8,9,10,11,12,13,14,16,17,18,19,20,21,22,23,24,25,26,27]. There are some other types of family proteins translated in yeasts [19]. TSC-22 cloned in our laboratory corresponds to TSC22D1-2, which is composed of 144 amino acids translated from a short variant of the TSC22D1 gene (Figure 1A). A protein of 86 amino acids is translated from the same short mRNA from the TSC22D1 gene and is referred to as TSC22(86) (TSC22D1-3) [17,19,20]. TSC22(86) (TSC22D1-3) does not have an NES and its structure has only a partial TSC-box and a leucine zipper (LZ). Most TSC-22 family proteins, including TSC-22 (TSC22D1-2), possess a TSC-box and LZ [8,9,10,11,12,13]. Kester et al. found that only leucine residues at positions 91 and 97 of the LZ domain are essential for TSC-22 family proteins (THG1) to form a homo- or heterodimer [24]. In addition, some TSC-22 proteins may interact with proteins that are not in the TSC-22 family or have no LZ [15]. However, regulatory factors upstream of TSC22D1 family proteins and downstream of intracellular signaling molecules that might bind to TSC22D1 family proteins have yet to be identified.

In the TSC-22 family, the function of TSC-22 (TSC22D1-2) has been widely studied [16,18,19,20,24,28,29,30,31,32,33,34,35,36,37]. The expression of TSC-22 (TSC22D1-2) is increased by various growth factors and hormones (TGF-β, FSH, TNF-α, EGF, FGF2) or cellular stress (activation of the RAS/MAP kinase pathway), and the TSC-22 (TSC22D1-2) protein induces differentiation, inhibits proliferation, and induces apoptosis in various cells [8,9,10,11,12,13,14,35,38,39] (Figure 1B). In addition, TSC22(86) (TSC22D1-3), which is upregulated by DNA damage, induces cellular senescence, such as senescence-associated heterochromatin foci (SAHF) formation and the senescence-associated secretary phenotype (SASP) [20]. To clarify the underlying molecular mechanisms of the action of the TSC22D1 family proteins, it is important to determine the localization of the TSC22D1 family proteins and to identify binding partners. In this study, we investigated the intracellular localizations of the TSC22D1 family proteins and attempted to identify the TSC22D1 family-binding proteins in each cellular location by mass spectrometry (MS).

## 2. Results

### 2.1. Identification of TSC-22 (TSC22D1-2)-Binding Protein by In Vitro GST Pull Down

In a GST-TSC-22 pull down assay using TYS cell extract, multiple binding proteins were found by electrophoresis (Figure 2). Each band was analyzed by mass spectrometry, and these proteins were identified as Elongation factor 1-gamma, Keratin, type II cytoskeletal 1, Carbonyl reductase [NADPH] 1, Pleckstrin homology domain-containing family G member 2, Rho-related GTP-binding protein Rho6, and Heat shock 70-kDa protein 1A/1B (Table 2 and Appendix A).

### 2.2. Intracellular Localization of Green Fluorescent Protein (GFP)-TSC-22 Fusion Proteins and Alterations of Location with DNA Damage

After transfection-induced overexpression of GFP-TSC-22 fusion proteins, TSC-22 (TSC22D1-2) (Figure 3) was found in the cytoplasm (Figure 4A). This was due to the NES located in the N-terminal region. The protein translocated into the nucleus after DNA damage (double-strand breakage) induced by phleomycin (Figure 4B). These results are consistent with our previous findings [10]. TSC-22N (Figure 3) retains the NES and was also localized in the cytoplasm (Figure 4C). However, TSC-22C (TSC22(86)) (Figure 3) lacks the NES and was present in both the cytoplasm and nucleus (Figure 4D).

### 2.3. Intracellular Localization of Endogenous TSC22D1 Proteins

Localization of endogenous TSC22D1 proteins was investigated in HEK293 cells by an immunofluorescence method using an anti-TSC22D1 antibody. Fluorescence appeared as dots in the cytoplasm, and nuclear fluorescence was also observed (Figure 5A). The location of a mitochondrial marker, Tim50 (Figure 5B), merged with that of TSC22D1 (Figure 5C). The consistency of the localized regions showed that cytoplasmic endogenous TSC22D1 is mostly present in the mitochondria. To clarify the intracellular localization of TSC22D1, Western blotting using mitochondrial and nuclear fractions of HEK293 cells was performed. This showed that endogenous TSC22D1-2 is mainly localized in the mitochondria (based on Tom20 expression) and nucleus (based on Lamin B1 expression) and that endogenous TSC22D1-1 is mainly localized in the nucleus (Figure 5D).

### 2.4. Identification of Intracellular TSC-22-Binding Proteins before and after DNA Damage

Flp-In^TM^T-REx^TM^293 cells were transfected with plasmids carrying TSC-22 (TSC22D1-2) (pcDNA5/FRT-Flag-TSC-22) or TSC-22C (TSC22(86)) (pcDNA5/FRT-Flag-TSC22(86)). In the TSC-22-transfected cells, binding proteins were found in the nuclear fraction after DNA damage (Figure 6A). Lamin-B1, Heat shock 70-kDa protein 1A/1B, Guanine nucleotide-binding protein-like 3 (GNL3), Pituitary adenylate cyclase-activating polypeptide type I receptor, Charged multivesicular body protein 4a, Histone H1.2, Heterogeneous nuclear ribonucleoprotein A1, and rRNA 2′-O-methyltransferase fibrillarin were identified by MS (Table 2). In TSC22(86)-transfected cells, Keratin, type II cytoskeletal 1, Ubiquitin carboxyl-terminal hydrolase 47, Protein tyrosine phosphatase receptor type C-associated protein, Keratin, type I cytoskeletal 9, Histone H1.4, Histone H1t, Histone H1.2, Actin, cytoplasmic 1, Glyceraldehyde-3-phosphate dehydrogenase, Dermcidin, 40S ribosomal protein S6, and 40S ribosomal protein S2 were identified as binding proteins in the whole cell extract after DNA damage (Figure 6B, Table 2, Appendix A).

### 2.5. Intracellular Binding of TSC-22 (TSC22D1-2) with Histone H1 or GNL3

Several candidates for the TSC-22-binding protein were identified in in vitro pull down assays or in vivo binding assays (Table 2). Among these proteins, we focused on histone H1, which is the main protein of chromatin in the nucleus and is closely involved in the control of gene expression, and GNL3, a nuclear protein known to interact with p53, a major tumor suppressor gene product. Histone H1.2 and GNL3 were also identified with high reliable values (Mascot score, peptide match number) by Mascot software, “92, 8” and “36, 4”, respectively. With the use of nuclear extract after DNA damage from Flp-In^TM^T-REx^TM^293 cells expressing Flag-TSC-22 (TSC22D1-2), immunoprecipitation was performed with anti-Flag antibody, followed by Western blotting with anti-histone H1 or anti-GNL3 antibody. Histone H1 was clearly detected, but GNL3 was not detectable (Figure 7). In a similar study using the whole cell extract of cells expressing Flag-TSC-22C (TSC22(86)), histone H1 was also detected (Figure 7), but GNL3 was not detectable (data not shown). In Western blotting of samples from GST-TSC-22 and GST-TSC-22C pull-down assays with anti-histone H1 and anti-GNL3 antibodies, histone H1 was not detectable, but GNL3 was detected in the sample of full-length TSC-22 (Appendix A).

## 3. Discussion

In this study, multiple candidates of binding proteins with the TSC22D1 family were identified in in vitro pull-down assays and in vivo binding assays (Table 2). Further analysis is needed to investigate whether these proteins actually bind to endogenous TSC22D1 family proteins under intracellular physiological conditions and to examine functional interactions, but it was clear that histone H1 binds to TSC-22 (TSC22D1-2) or TSC22(86) (TSC22D1-3) in cells. Histone H1 is a linker histone and its overexpression induces an abnormal nuclear morphology and chromatin structure, which positively and negatively regulates transcription in vitro (Figure 8A) [20,40,41,42,43]. We propose the possible mechanisms between the TSC22D1 family and histone H1 (Figure 8A). The TSC22D1 family may promote the degradation of histone H1 through this interaction, and replacement of chromatin binding by HMGA2 may cause changes in the expression of genes that regulate cell proliferation and differentiation, apoptosis induction, and cell senescence. GNL3 was identified as a TSC-22 (TSC22D1-2)-binding protein candidate in the transfected cells. This protein is also known as nucleostemin and is found in the nucleolus. GNL3 interacts with p53 and MDM2 and thus influences the cell cycle progression and cellular differentiation, with a decrease in GNL3 (Figure 8B) [44,45]. GNL3 also inhibits proliferation of several tumor cells [46,47].

The binding proteins identified in the in vitro pull down assay were not always consistent with those found in the in vivo binding assay (Table 2). In TSC-22-transfected cells, TSC-22 (TSC22D1-2) was shown to bind to histone H1, but not to GNL3. In contrast, GNL3 was detected in the pull down assay, but histone H1 was not detectable. The reasons for the discrepancies are unclear, but GST (molecular weight: 26 kDa) and Flag (1 kDa) may influence the conformation of TSC-22 (16 kDa) or its interaction with other proteins.

TSC-22 (TSC22D1-2) has a N-terminus NES and is localized in the cytoplasm in a static state but translocates into the nucleus under stress such as cytotoxicity [10]. We showed that an artificially produced mutant (TSC-22LZ) without the NES is localized in both the cytoplasm and nucleus [11]. A very small protein naturally produced from the TSC22D1 gene is structurally similar to TSC-22LZ, and this protein was later named as TSC22(86) (TSC22D1-3) [19]. In this study, we used TSC22(86) (TSC22D1-3) but not TSC-22LZ as a TSC22D1 family protein without NES. Given that identification of binding proteins after DNA damage is needed to clarify the function of TSC22D1 family proteins, an analysis of TSC-22-binding proteins was performed using nuclear extract from cells expressing TSC-22 (TSC22D1-2) and in whole cell extract from cells expressing TSC22(86), both after loading stress by DNA double-strand breakage induced by phleomycin treatment. In both analyses, many binding proteins were observed after DNA damage, indicating that the TSC22D1 protein family may bind to more proteins in the presence of stress.

To understand the role of TSC-22 in intracellular signaling, a detailed determination of the intracellular localizations of TSC22D1 family proteins is needed. We could not determine the specific cytoplasmic regions or organelles in which TSC-22 localized in the transfection assay because the expression level of transfected TSC-22 was too high. Localization of endogenous TSC22D1 family proteins in HEK293 cells was determined by immunofluorescence using anti-TSC22D1 antibody. Fluorescent dots in the cytoplasm and nuclear fluorescence were observed. Similar results for the localization of endogenous TSC22D1 family proteins were obtained in other cells (U2OS, RH30, MCF7, and HOS) (Appendix A). The antibody used is a polyclonal antibody against the internal region of the TSD22D1 protein and reacts with all TSC22D1 isoforms (Table 1) [34,37,48]. These proteins are mainly TSC22D1-1 and TSC22D1-2 [37]. Western blotting showed that TSC22D1-1 was mostly in the nucleus and TSC22D1-2 was in the mitochondria and nucleus. We tried to find a mitochondria-targeting signal (MTS) in TSC-22 (TSC22D1-2), but we could not find an MTS-like motif in TSC-22 (TSC22D1-2). It is reported that there is no core MTS sequence, and the pre-sequences showing soft interactions with mitochondrial outer membrane proteins, such as Tom20, are diverse. Among the mitochondria proteins, SSBP1 is reported to translocate to the nucleus by a carrier protein HSF1 under stress and to then regulate the transcription of several genes [49]. TSC-22 (TSC22D1-2) might have a similar function. Recently, we identified C1QBP as a TSC-22 (TSC22D1-2)-binding protein candidate localized in the mitochondria in a preliminary experiment (unpublished data), and we are currently investigating the details of its interaction.

Although we need to confirm whether the candidates for the TSC22D1 family-binding nuclear protein really bind to endogenous TSC22D1 family proteins under physiological conditions, further investigation of the molecular interaction of the binding proteins with TSC22D1 family proteins would clarify the biological roles of the TSC22D1 family protein in several cell systems.

## 4. Material and Methods

### 4.1. Cell Lines and Cell Culture

TYS cells [50] and HEK293 cells were used in the study. TYS cells were obtained from Dr. Yanagawa at the Second Department of Oral and Maxillofacial Surgery, Tokushima University School of Dentistry. TYS cells are human oral adenosquamous carcinoma cells that were previously established by our group. HEK293 cells are derived from human embryonic kidney and are widely used for transfection experiments. The HEK293 cells used in this study were purchased from JCRB (NIBIO, Osaka, Japan) and maintained at the Division of Proteomics, Institute of Development, Aging, and Cancer, Tohoku University. For the Flp-In^TM^ system, we used Flp-In^TM^293 T-Rex cells (Thermo Fisher Scientific, Waltham, MA, USA). These cells contain a single stably integrated flippase (Flp) recognition target (FRT) site at a transcriptionally active genomic locus and stably express the Flp recombinase from pOG44 [51,52]. The cells were cultured in 100 mm culture dishes and maintained with Dulbecco’s modified Eagle’s medium (DMEM; Merck Sigma–Aldrich, St. Louis, MO, USA) supplemented with 10% fetal bovine serum (FBS; Thermo Fisher Scientific) and 5% antibiotic-antimycotic solution (Fujifilm, Wako Pure Chemical, Osaka, Japan). In all experiments, cells were incubated at 37 °C in a humidified 5% CO_2_ atmosphere. The origin and identity of the TYS cells used in the study were previously confirmed using short tandem repeat analysis [53].

### 4.2. Construction of the Expression Vector

#### 4.2.1. pEGFP-TSC-22 Fusion Protein Expression Vector

Using a plasmid pEGFP-TSC-22 [10] containing human TSC-22 (TSC22D1-2) cDNA, various truncation mutants were made using the primers below. pEGFP-TSC-22N encodes a protein (that does not exist naturally) with only the N-terminal region (amino acids 1-60) of human TSC-22(TSC22D1-2). pEGFP-TSC-22C encodes TSC22(86) (TSC22D1-3) (Figure 3). The primers used were as follows: TSC22D1-Xho I (35 mer): ATC TCG AGA TGC ACC AGC CGC CTG AGT CCA CCG CC; TSC22D1 Not I (35 mer): ATG CGG CCG CAC TAT GCG GTT GGT CCT GAG CCC TG; TSC22-Xho I (27 mer): ATC TCG AGA TGA AAT CCC AAT GGT GTA; TSC22-Not I (29 mer): ATG CGG CCG CAT GCG GTT GGT CCT GAG CC; TSC22-M39-Xho I (26 mer): ATC TCG AGA TGG ATC TAG TGA AAA GC; TSC22-D60-Not I (29 mer): ATG CGG CCG CAA TCC ATA GCT TGC TCG AT.

#### 4.2.2. Flag-TSC-22 and Flag-TSC22(86) Fusion Protein Expression Vectors

To make Flp-In^TM^293 T-Rex cells expressing TSC-22 proteins using the Flp-In^TM^ system, two expression vectors were prepared. pcDNA5/FRT-Flag-TSC-22 contained full-length human TSC-22 (TSC22D1-2) fused with the Flag protein gene at the pcDNA5/FRT cloning site. pcDNA5/FRT-Flag-TSC22(86) contained TSC22(86) fused with the Flag protein gene at the pcDNA5/FRT cloning site, in which the N-terminal 58 amino acids of TSC-22 (TSC22D1-2) were deleted. Since the region in which the exogenous gene is integrated in the genome is defined beforehand in the Flp-In^TM^ system, expression of the introduced gene is strictly controlled, and the gene transfer system is unlikely to influence or be influenced by the genome [51,52].

### 4.3. Determination of the TSC-22 (TSC22D1-2)-Binding Protein by In Vitro GST Pull Down

In a previous study, we constructed a plasmid vector (pGEX4T-2-TSC-22) expressing glutathione S-transferase (GST) and the full-length human TSC-22 (TSC22D1-2) fusion protein in *E. coli* [9]. The *E. coli* BL21 cell line was transformed with pGEX4T-2-TSC-22 and pre-cultured in 15 mL of LB medium containing ampicillin (0.1 mM) (both Funakoshi, Tokyo, Japan) at 26 °C for 16 h. Then, the culture was transferred into 150 mL of LB medium containing ampicillin (0.1 mM), combined with 0.1 mM isopropyl β-D-thiogalactopyranoside (IPTG, Funakoshi), and cultured at 37 °C for 4 h. The bacteria were then collected. One mL of the extract of *E. coli* expressing GST-TSC-22 was incubated with 50 mL of Glutathione Sepharose^TM^ 4B (GE Healthcare, IL, USA) at 4 °C for 1 h, washed once with washing buffer A [50 mM Tris-HCl (pH 7.2), 150 mM NaCl, 0.2% Nonidet P-40 (NP-40)], washed once with washing buffer B [50 mM Tris-HCl (pH 7.2), 1.2 M NaCl)], and finally washed once with Ca^2+^- and Mg^2+^-free Dulbecco’s Phosphate-Buffered Saline (D-PBS(-)) (all Fujifilm, Wako Pure Chemical). In parallel, the cell pellet collected from subconfluent conditions in one 100 mm culture dish was combined with 6 mL of extraction buffer A [50 mM Tris-HCl (pH 7.5), 350 mM NaCl, 1 mM 2-mercaptoethanol (2ME; Fujifilm, Wako Pure Chemical), 0.1% NP-40)], sonicated on ice for 10 s twice, and centrifuged at 18,000× *g* for 10 min, and the supernatant was collected as the cell extract. GST-TSC-22-conjugated glutathione beads and 1.5 mL of the adjusted cell extract were mixed and incubated at 4 °C for 6–12 h. The glutathione beads were washed twice with washing buffer C [50 mM HEPES (pH 7.2) (Fujifilm, Wako Pure Chemical), 150 mM NaCl, 0.2% NP-40] and once with D-PBS(-). The binding proteins were collected from the glutathione beads with 0.5 mL of extraction buffer B [50 mM HEPES (pH7.2), 1.2 M NaCl)] twice and subjected to desalting and concentration to 50 μL using a MW10,000 cut-off ultracentrifugal concentrator (Amicon Ultra-4, Millipore, Burlington, MA, USA). To 5 μL of this sample, 5 μL of 2×SDS-PAGE Sample buffer [62.5 mM Tris-HCl (pH 6.8), 2% sodium dodecyl sulfate (SDS), 25% glycine, 0.01% bromophenol blue, Thermo Fisher Scientific] was added, and the mixture was incubated at 98 °C for 3 min and used as the electrophoresis sample. The sample (10 μL) was applied to each lane of a gel (Mini-Protein-TGX, 4–20%, Bio-Rad, Hercules, CA, USA) and electrophoresed at 200 V constant voltage for 30 min for separation (SDS-PAGE electrophoresis buffer: 250 mM Tris-HCl, 1.92 M glycine, 1% SDS; Nippon GENE, Toyama, Japan). Binding proteins were detected by silver staining using a silver stain kit for mass spectrometry (Fujifilm, Wako Pure Chemical).

### 4.4. Mass Spectrometry

Protein bands separated by SDS-PAGE were cut out, decolorized, and dehydrated with acetonitrile (Fujifilm, Wako Pure Chemical). The gel fragments were reduced and alkylated with 100 mM dithiothreitol and 100 mM iodoacetamide (both Merck Sigma–Aldrich). After washing with ultrapure water, the gel fragments were treated with trypsin (Fujifilm, Wako Pure Chemical) at 30 °C for 12 h and desalted using Ziptip c18 (Merck, Millipore). The sample was analyzed using a nanoLC/MS/MS system (DiNa HPLC, Kya Tech Corp./QSTAR XL Applied Biosystems, Tokyo, Japan). MS ion data were analyzed using Mascot software version 2.5.1 (Matrix Science, Tokyo, Japan) against the SwissProt database with the setting of 1 mis-cleavage and the fixed modification: Oxidation (Methionine), Carbamidomethyl (Cysteine).

### 4.5. Transfection

TYS or HEK293 cells were transfected with pEGFP-TSC-22, pEGFP-TSC-22N, or pEGFP-TSC22(86) using Lipofectamine 3000, followed by selection with 200 mg/mL G418 (both Thermo Fisher Scientific). Cells expressing GFP-TSC-22, GFP-TSC-22N, or GFP-TSC22(86) were used in bulk without cloning in the following experiments. Flp-In^TM^T-REx^TM^293 cells were transfected with full-length TSC-22 (pcDNA5/FRT-Flag-TSC-22) or TSC22(86) (pcDNA5/FRT-Flag-TSC22(86)) using Lipofectamine 3000, followed by selection with 50 mg/mL hygromycin B (Fujifilm, Wako Pure Chemical). Cells expressing the plasmid were also used in bulk without cloning.

### 4.6. Intracellular Localization of the TSC-22 Fusion Protein

HEK293 cells expressing GFP-TSC-22 (full-length) were treated with 0.5 mM phleomycin (Funakoshi) for 12 h, and localization of GFP-TSC-22 was examined before and after DNA damage. In HEK293 cells transfected with GFP-TSC-22N and GFP-TSC22(86), protein localization was examined without DNA damage. In observation of fluorescent proteins, cells cultured in a 1.5-cm glass bottom dish (AS ONE, Osaka, Japan) were washed with D-PBS(-), and GFP fluorescence was observed by confocal fluorescence microscopy at an excitation wavelength of 488 nm.

### 4.7. Localization of Endogenous TSC-22

Localization of endogenous TSC-22 was examined in HEK293 cells. The cells were cultured in a 1.5-cm glass-bottom dish to 80% confluence, the medium was removed, and the cells were washed twice with D-PBS(-) and fixed in 10% acetaldehyde (Fujifilm, Wako Pure Chemical)-containing D-PBS(-) at room temperature for 10 min. After washing once with D-PBS(-), the sample was incubated in 1% Triton X-10 (Merck Sigma–Aldrich)-containing D-PBS(-) at room temperature for 10 min, washed twice with D-PBS(-), and then blocked with 1% Blockace (Megmilk Snow Brand, Hokkaido, Japan) at room temperature for 1 h. After blocking, the sample was washed with D-PBS(-) and then incubated with rabbit anti-human TSC22D 1 polyclonal antibody (Proteintech, Tokyo, Japan) or mouse anti-human Tim50 monoclonal antibody (Santa Cruz Biotechnology, Dallas, TX, USA), each was 100-fold diluted with Can get signal immunostain solution A (Toyobo, Osaka, Japan) at room temperature for 1 h. After washing three times with TTBS (50 mM Tris-HCl (7.6), 150 mM NaCl, 0.05% Tween 20; Takara Bio, Shiga, Japan) for 5 min, the sample was incubated with Alexa Fluor secondary antibody (FITC-conjugated anti-rabbit antibody or Texas red conjugated anti-mouse antibody) (Thermo Fisher Scientific) and 200-fold diluted with 1% Blockace at room temperature for 1 h. After washing five times with TTBS, the sample was mounted with DABCO (Merck Sigma Aldrich), and Texas red and FITC fluorescence were observed by confocal fluorescence microscopy at excitation wavelengths of 495 and 590 nm, respectively.

### 4.8. Identification of Binding Proteins before and after DNA Damage

Flp-In^TM^T-Rex^TM^293 cells expressing Flag-TSC-22 and Flag-TSC22(86) were used after treatment with 0.5 mM phleomycin for 12 h with or without treatment. The cells were collected in a 1.5-mL Eppendorf tube using a cell scraper (AGC Techno Glass, Shizuoka, Japan) and washed with D-PBS(-). The cell pellet was combined with 1.0 mL of SHE buffer (10 mM HEPES (pH 7.5), 210 mM mannitol, 7 mM sucrose, 1 mM EDTA, 1 mM EGTA, 0.15 mM spermine, 0.75 mM spermidine) (Fujifilm, Wako Pure Chemical) and homogenized on ice using a Potter-type homogenizer by applying 10 strokes to disrupt the cells. The disrupted cell suspension was centrifuged at 700× *g* for 10 min to separate into the supernatant and precipitate, which were regarded as the cytoplasmic and crude nuclear fractions, respectively. The crude nuclear fraction was resuspended with SHE buffer and centrifuged at 18,000× *g* for 10 min to wash the precipitate. The precipitate was combined with 1 mL of cell extraction buffer C [50 mM HEPES (pH 7.5), 350 mM NaCl, 1 mM centrifuged 2ME, 0.1% NP-40, Protease Inhibitor Cocktail (Thermo Fisher Scientific), Benzonase (Santa Cruz Biotechnology)], sonicated for 10 sec, and centrifuged at 18,000× *g* for 10 min, and the supernatant was regarded as the nuclear fraction. To 1 mL of the nuclear or whole cell extract, 30 μL of anti-FLAG M2-Agarose beads (Merck Sigma–Aldrich) was added and incubated at 4 °C for 4 h, followed by centrifugation at 1000× *g* for 2 min, and the supernatant was discarded. The remaining beads were washed twice with washing buffer (50 mM HEPES (pH 7.5), 150 mM NaCl, 0.1% NP-40) and once with D-PBS(-). After combination with 30 μL of Elution buffer (20 mM Tris-HCl (pH 2.5), 100 mM glycine) and standing on ice for 1 min, the beads were centrifuged at 18,000× *g* for 2 min and the supernatant was collected. After neutralizing by adding 2 μL of 1 M Tris-HCl (pH 9.5) and adding the same volume of 2×SDS-Sample buffer, the sample was boiled at 98 °C for 4 min. The cytoplasmic and nuclear fractions were individually electrophoresed as described above, and the gel was subjected to silver staining. The detected bands were cut out and analyzed by mass spectrometry.

### 4.9. Immunoprecipitation-Western Blotting

The nuclear fraction of Flp-In^TM^T-Rex^TM^293 cells expressing Flag-TSC-22 was prepared. A whole-cell extract was prepared from Flp-In^TM^T-Rex^TM^293 cells expressing Flag-TSC22(86). The nuclear or whole-cell extracts were immunoprecipitated using anti-FLAG M2-Agarose and electrophoresed on a 4–20% gradient gel (Multigel II mini 4/20, Cosmo Bio, Tokyo, Japan). After electrophoresis, the blot was prepared by applying an electric current for 7 min using a Bio-Rad Trans-Blot Turbo transfer system and blocked with 2% bovine serum albumin (BSA, Fujifilm, Wako Pure Chemical)-TTBS for 1 h. The blotted membrane was incubated with rabbit anti-human TSC22D1 polyclonal antibody, mouse anti-human histone H1 monoclonal antibody (Genetex, Irvine, CA, USA), and rabbit anti-human GNL3 polyclonal antibody (Genetex) 1000-fold diluted with 2% BSA-TTBS at room temperature for 2 h. Then, washing with TTBS for 10 min was repeated three times, and the blots were incubated with 4000-fold diluted HRP-conjugated anti-rabbit or anti-mouse secondary antibody (Santa Cruz Biotechnology) at room temperature for 1 h, followed by washing with TTBS for 10 min three times and detection using ELC Plus (Thermo Fisher Scientific) and X-ray film (Fuji super RX, Fujifilm, Tokyo, Japan).

### 4.10. Western Blotting Using Mitochondrial and Nuclear Fractions

Nuclear or mitochondrial extract was prepared from HEK293 cells. The cell pellet was suspended in 1 mL of SHE buffer and homogenized using a Potter-type homogenizer on ice by applying 10 strokes to disrupt the cells. The disrupted cell suspension was centrifuged at 700× *g* for 10 min to separate the supernatant and precipitate, and the precipitate was regarded as the crude nuclear fraction. The supernatant was further centrifuged at 18,000× *g* for 10 min, and the supernatant and precipitate were regarded as the cytoplasmic and crude mitochondrial fractions, respectively. The crude mitochondrial and crude nuclear fractions were resuspended with SHE buffer and centrifuged at 18,000× *g* for 10 min to wash the precipitate of each fraction. To each precipitate, 1 mL of cell extraction buffer C was added. The precipitates were sonicated for 10 sec and centrifuged at 18,000× *g* for 10 min, and the supernatants were defined as the mitochondrial and nuclear fractions. The purity of each fraction was shown by Western blotting using rabbit anti-human Tom20 polyclonal antibody (Santa Cruz Biotechnology) and rabbit anti-human LaminB1 polyclonal antibody (Abcam, Cambridge, UK) as markers of the mitochondrial and nuclear fractions, respectively.

## Figures and Tables

**Figure 1 ijms-22-10913-f001:**
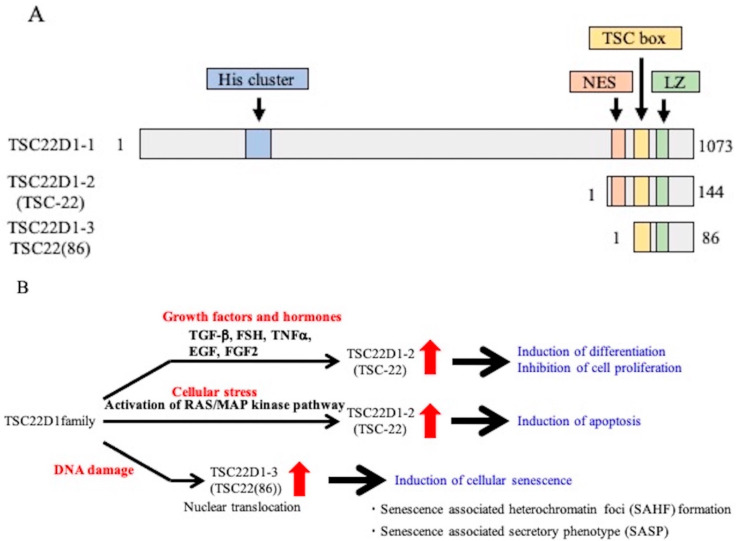
Structure and function of TSC22D1 family proteins. (**A**): Structures of TSC22D1-1, TSC22D1-2 (TSC-22), and TSC22D1-3 (TSC22(86)). TSC22D1-1 has 1,073 amino acids, a His cluster in the N-terminal region, and a nuclear export signal (NES), TSC-box, and leucine zipper (LZ) close to the C-terminus. TSC22D1-2 has 144 amino acids of the C-terminal of TSC22D1-1 and an NES in the N-terminal region and a TSC-box and LZ close to the C-terminus. TSC22D1-3 has 86 amino acids of the C-terminal of TSC22D1-2, lacks an NES, but has a TSC-box and LZ. (**B**): Functions of the TSC22D1 family. Expression of TSC-22 (TSC22D1-2) is increased by growth factors; the protein induces cell differentiation, inhibits cell proliferation, and induces apoptosis. TSC22(86) (TSC22D1-3) induces cell senescence.

**Figure 2 ijms-22-10913-f002:**
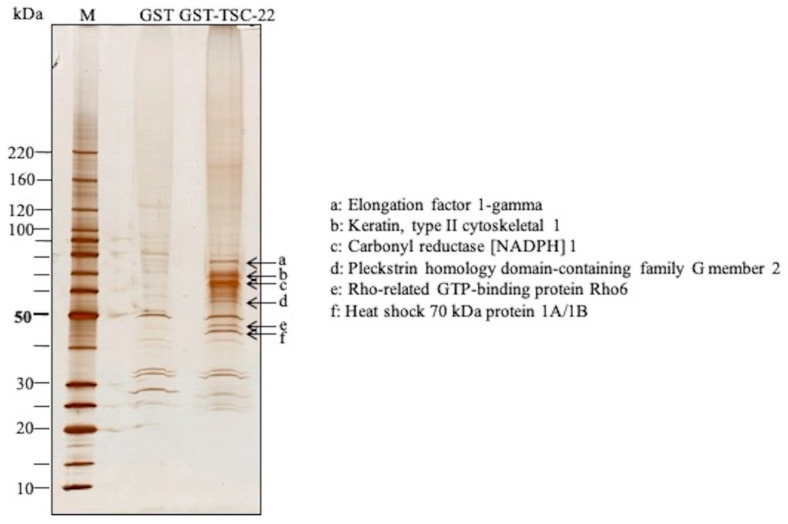
Identification of TSC-22 (TSC22D1-2)-binding proteins by in vitro GST pull down. ←: TSC-22-binding protein. a: Elongation factor 1-gamma, b: Keratin, type II cytoskeletal 1, c: Carbonyl reductase [NADPH] 1, d: Pleckstrin homology domain-containing family G member 2, e: Rho-related GTP-binding protein Rho6, f: Heat shock 70-kDa protein 1A/1B.

**Figure 3 ijms-22-10913-f003:**
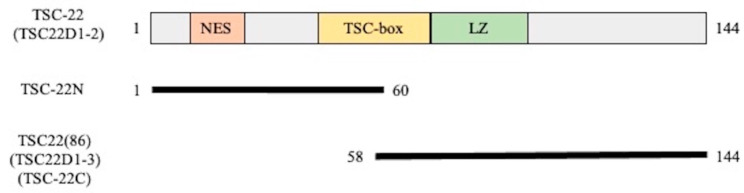
Structure of TSC-22 (TSC22D1-2) and truncation mutants used in this study. NES: nuclear export signal, LZ: leucine zipper. TSC-22N is not a naturally occurring protein.

**Figure 4 ijms-22-10913-f004:**
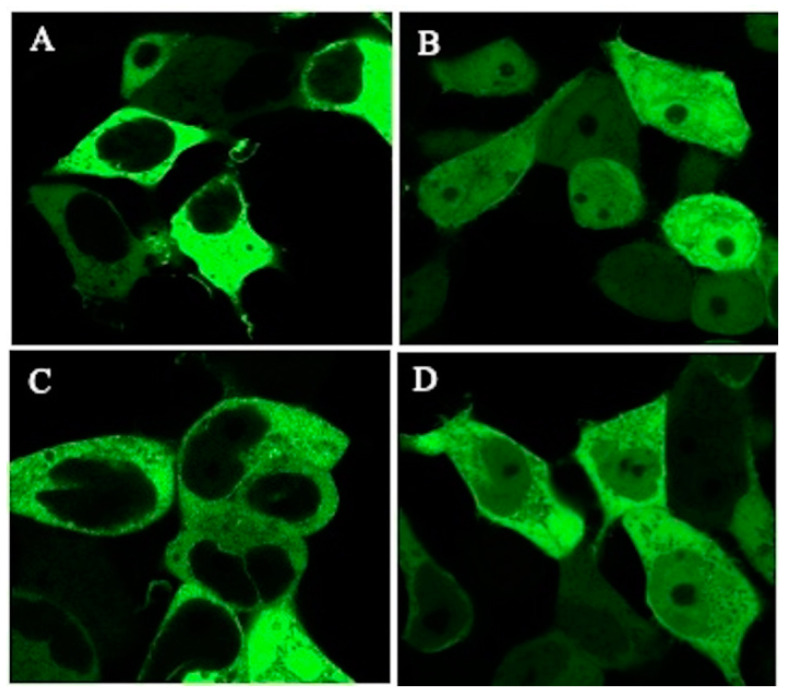
Intracellular localization of transfected TSC-22 (TSC22D1-2) and truncation mutants (TSC-22N, TSC-22C (TSC22(86)). (**A**,**B**): TSC-22 under (**A**) unstimulated conditions and (**B**) after treatment with phleomycin (0. 5 mM) for 12 h. (**C**): TSC-22N. (**D**): TSC-22C (TSC22(86)).

**Figure 5 ijms-22-10913-f005:**
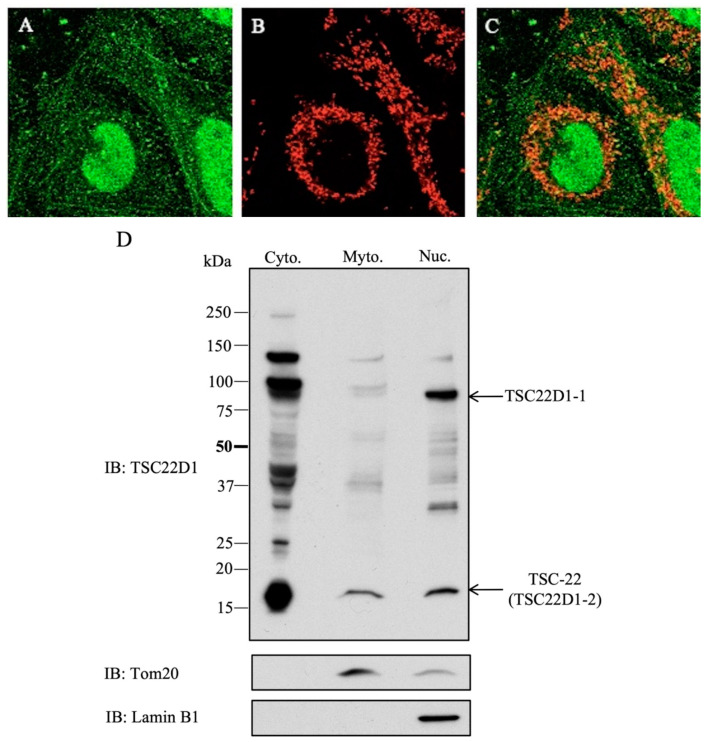
Intracellular localization of endogenous TSC22D1 proteins. (**A**): Fluorescence antibody method using anti-TSC22D1 antibody. HEK293 cells showed fluorescence as dots in the cytoplasm and nuclear fluorescence due to endogenous TSC22. (**B**): Localization of a mitochondrial marker, Tim50. (**C**): Merging of TSC22D1 and TIM50. (**D**): Western blotting using mitochondrial and nuclear fractions of HEK293 cells.

**Figure 6 ijms-22-10913-f006:**
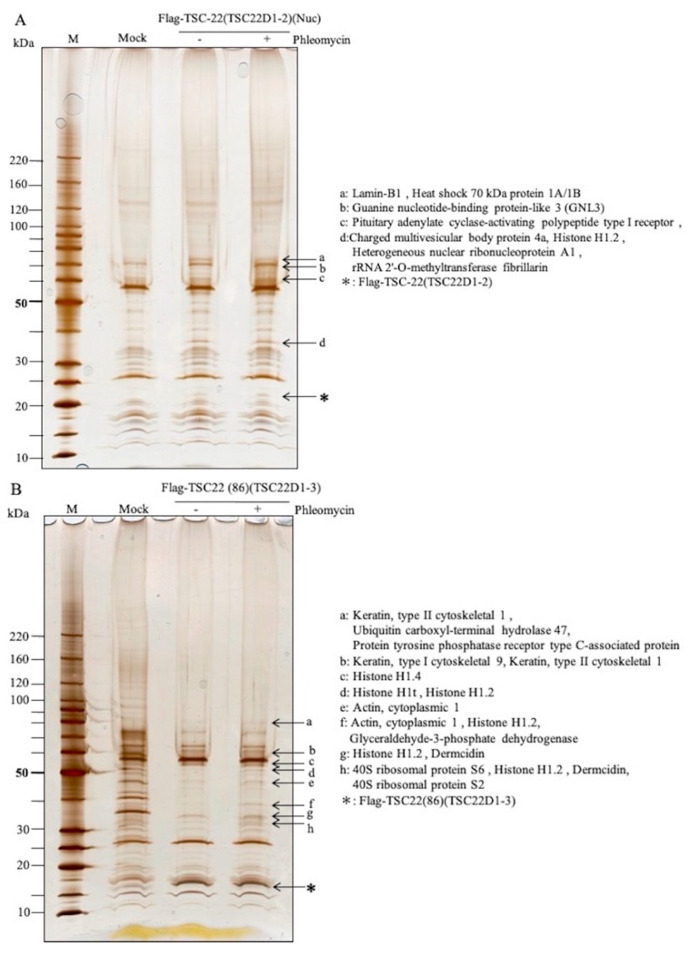
Identification of TSC-22-binding proteins in cells before and after DNA damage. (**A**): TSC-22-binding protein (arrow) in the nuclear fraction after DNA damage by phleomycin in cells expressing full-length TSC-22. a: Lamin-B1, Heat shock 70-kDa protein 1A/1B, b: Guanine nucleotide-binding protein-like 3 (GNL3), c: Pituitary adenylate cyclase-activating polypeptide type I receptor, d: Charged multivesicular body protein 4a, Histone H1.2, Heterogeneous nuclear ribonucleoprotein A1, rRNA 2′-O-methyltransferase fibrillarin, *Flag-TSC-22. (**B**): TSC22(86)-binding proteins (←) after DNA damage by phleomycin in a whole cell fraction of cells expressing TSC22(86). a: Keratin, type II cytoskeletal 1, Ubiquitin carboxyl-terminal hydrolase 47, Protein tyrosine phosphatase receptor type C-associated protein, b: Keratin, type I cytoskeletal 9, Keratin, type II cytoskeletal 1, c: Histone H1.4, d: Histone H1t, Histone H1.2, e: Actin, cytoplasmic 1, f: Actin, cytoplasmic 1, Histone H1.2, Glyceraldehyde-3-phosphate dehydrogenase, g: Histone H1.2, Dermcidin, h: 40S ribosomal protein S6, Histone H1.2, Dermcidin, 40S ribosomal protein S2, *Flag-TSC22(86).

**Figure 7 ijms-22-10913-f007:**
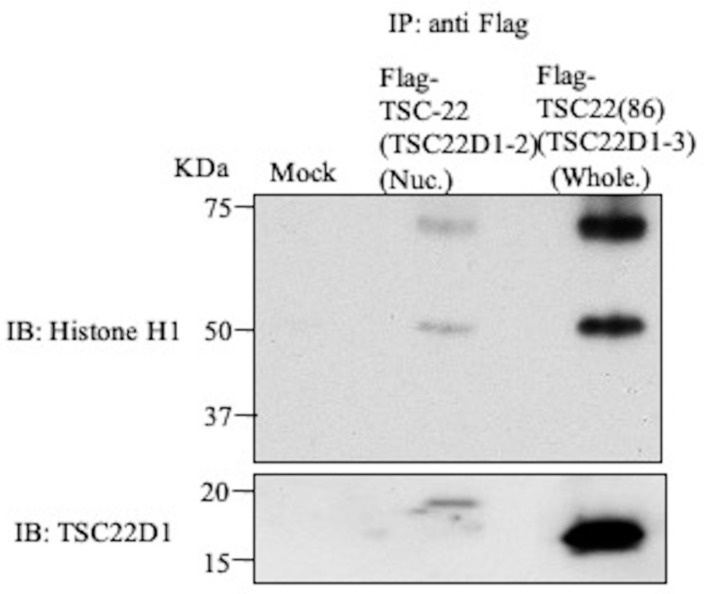
Intracellular binding of TSC-22 (TSC22D1-2) with histone H1. IP-Western blotting using nuclear extract after DNA damage in Flag-TSC-22 (TSC22D1-2)-transfected cells. In detection using anti-histone H1 or anti-GNL3 antibody, histone H1 was clearly detected.

**Figure 8 ijms-22-10913-f008:**
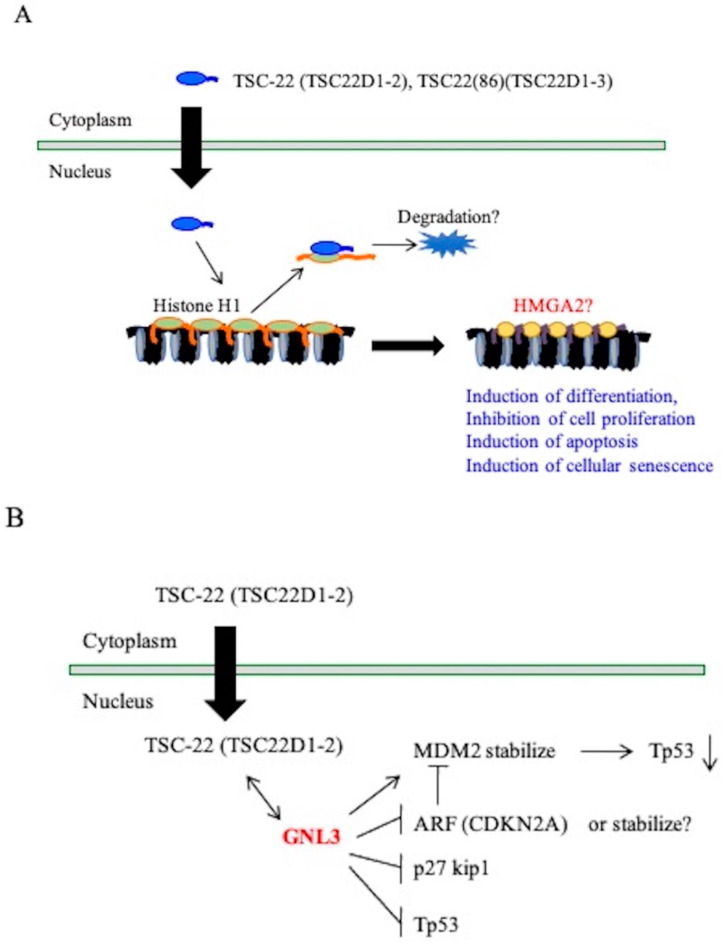
Putative function of TSC-22 (TSC22D1-2) and TSC-22(86) (TSC22D1-3). (**A**): TSC-22 may promote the degradation of histone H1 through this interaction, and replacement of chromatin binding by HMGA2 may cause changes in the expression of genes that regulate cell proliferation and differentiation, apoptosis induction, and cell senescence. (**B**): GNL3 is known as nucleostemin and is found in the nucleolus. GNL3 interacts with p53 and MDM2. TSC-22 (TSC22D1-2) might influence the inhibitory action of GNL3 on tumor cell growth.

**Table 1 ijms-22-10913-t001:** TSC-22 family.

Gene Name	Transcript or Protein Name (NCBI Accession Number)	Other Names	Protein Length (Number of Amino Acids)	Reference
TSC22D1	TSC22D1-1(NP_904358.2)	TSC22D1-X1	1073	[15,16,17]
	TSC22D1-2 (NP_006013.1)	TSC-22/TSC22D1-CRA_a	144	[8,9,10,11,12,13,14,15,17,18]
	TSC22D1-3 (NP_001230727.1)	TSC22 (86)	86	[17,19]
	TSC22D1-4 (NP_001230728.1)		570	[17,19,20]
	TSC22D1-X2 (XP_016876299.1)		1048	[17]
	TSC22D1-X3 (XP_005266640.1)		1028	[17]
	TSC22D1-X4 (XP_016876300.1)		985	[17]
TSC22D2	TSC22D2-1 (NP_055594.1)	TSC22D2-CRA_b	780	[15,16,17,19]
	TSC22D2-2 (NP_001290193.1)		756	[17,25]
	TSC22D2-X1 (XP_011511639.1)		691	[17]
	TSC22D2-CRA_a (EAW78837.1)		690	[17,26]
	TSC22D2-CRA_c (EAW78840.1)		753	[17,26]
	TSC22D2-CRA_d (EAW78841.1)		124	[17,26]
TSC22D3	TSC22D3-1 (NP_932174.1)	TSC22D2-X1 TSC22D2-CRA_a	200	[16,17,19,20]
	TSC22D3-2 (NP_004080.2)	GILZ/T SC22D2-CRA_c	134	[15,17,21,22]
	TSC22D3-3 (NP_001015881.1)		77	[17,27]
	TSC22D3-CRA_b (EAX02707.1)		193	[17,26]
	Long-GILZ (ACJ09091.1)		234	[17,22]
TSC22D4	TSC22D4-1 (NP_001289972.1)	THG-1 / TSC22D4-a	395	[17,23,24]
	TSC22D4-2		310	[15,19]
	TSC22D4-3		195	[15,19]

**Table 2 ijms-22-10913-t002:** Candidates of TSC-22 binding proteins identified by MS (UniProt Accession number).

	GST-TSC-22 Pull-Down Assay (TYS, Whole Cell Extracts)	In Vivo Flag-TSC-22 Binding Assay(HEK293, Nuclear Extracts)	In Vivo Flag-TSC22 (86) Binding Assay (HEK293, Whole Cell Extracts)
Chromatin structure		Histone H1.2 (P16403) Lamin-B1 (P20700)	Histone H1.2 (P16403), Histone H1.4 (P10412) Histone H1t (P22492)
Cell growth, Apoptosis, differentiation		Guanine nucleotide-binding protein-like 3 (Q9BVP2)	40S ribosomal protein S6 (P62753)
Signal transduction		Pituitary adenylate cyclase-activating polypeptide type I receptor (P41586)	
Chaperone	Heat shock 70-kDa protein 1A/1B (P0DMV8/P0DMV9)	Heat shock 70-kDa protein 1A/1B (P0DMV8/P0DMV9)	
Cytoskeleton	Keratin, type II cytoskeletal 1 (P04264) Rho-related GTP-binding protein Rho6 (Q92730) Pleckstrin homology domain-containing family G member 2 (Q9H7P9)		Keratin, type II cytoskeletal 1 (P04264) Keratin, type I cytoskeletal 9 (P35527) Actin, cytoplasmic 1 (P60709)
Others	Elongation factor 1-gamma (P26641) Carbonyl reductase [NADPH] 1 (P16152)	Heterogeneous nuclear ribonucleoprotein A1 (P09651) Charged multivesicular body protein 4a (Q9BY43) rRNA 2′-O-methyltransferase fibrillarin (P22087)	Glyceraldehyde-3-phosphate dehydrogenase (P04406) Ubiquitin carboxyl-terminal hydrolase 47 (Q96K76) Dermcidin (P81605) Protein tyrosine phosphatase receptor type C-associated protein (Q14761) 40S ribosomal protein S2 (P15880)

## Data Availability

Not applicable.

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
