# Peer review of "Identification of Binding Proteins for TSC22D1 Family Proteins Using Mass Spectrometry"

_ijms, 2021, doi:10.3390/ijms222010913_

Round 1
Reviewer 1 Report
The manuscript in its current form is not suitable for publication. Due to the lack of relevant information (see # 1) or the lack of precision of this information (see # 2), it is impossible to assess the reliability of the data presented and the conclusions drawn from this data.
From the content of the work, it is difficult to assess the relevance of the presented problem (see # 3).
Major issues
#1
The authors do not provide information on how reliable the protein identifications presented in Table 2 are. There are no standard data provided on the reliability of the identifications made (number of identified peptides, for each protein, coverage, FDR threshold value used to filter the results, etc.). Such data could be provided in supplemental data.
In the description of the methodology of proteomic measurements, there is no description of how data from measurements were processed and analyzed (software, databases, setup of search parameters etc.)
#2
Table 2 does not contain accurate information about the identified proteins.
- a) Most of the symbols used are symbols used to describe genes, e.g. RPS6, GNL3. The results of proteomic research refers to amino acid sequences (products of the gene) and names of proteins must be used.
- b) some of the descriptions are very imprecise, e.g. there are several genes of the HSP70 protein in humans (Cell Stress & Chaperones. 1 (1): 23–8. doi:10.1379/1466-), which protein or proteins the authors identified?
- c) there are at least several hundred receptor proteins containing seven transmebrane helixes in the human body? What proteins from this family have been identified?
Summing up, the data contained in Table 2 are to be precise by giving a reference to the protein sequence (and not the names of genes), it is not the genes that are being studied in this case.
Modify the data in Table 2 (unify the naming and use of protein sequence names/symbols). It must cause changes to the text and descriptions of the drawings so that uniform naming is applied throughout the text.
# 3
10 out of 43 (references 1-2, 8-14, 43) literature items are self-citation. It can be assumed that the topic presented in the manuscript is very niche.
Other issues
The first part of Introduction lines 41-50 is completely redundant to the readability of the manuscript and should be removed.
Lines 77-78 „Among the TSC-22 family, the function of TSC-22 (TSC22D1-2) has been widely studied” – here should be references to this extensive research ?
Table 1. For most transcripts, the only reference is to the NCBI. However, all these transcripts have been described in scientific publications and it is these publications that should be cited as literature references. For example, the transcript of TSC22D1-4 (NP_001230728.1) is described in the publication Genet Med. 2017 Oct;19(10):1105-1117. doi: 10.1038/gim.2017.37. Epub 2017 May 11. . In the description of the table, it must be stated that the accession numbers refers to the NCBI database (reference 17).
Where in Figure 7 should be the band for G protein of the nucleolar 3 protein (GLN3 gene) ?
Author Response
Response to Reviewer 1 Comments
The manuscript in its current form is not suitable for publication. Due to the lack of relevant information (see # 1) or the lack of precision of this information (see # 2), it is impossible to assess the reliability of the data presented and the conclusions drawn from this data. From the content of the work, it is difficult to assess the relevance of the presented problem (see # 3).
Response: Thank you very much for your critical reading and comments. We revised the manuscript according to the reviewer's suggestions as follows.
Point #1: The authors do not provide information on how reliable the protein identifications presented in Table 2 are. There are no standard data provided on the reliability of the identifications made (number of identified peptides, for each protein, coverage, FDR threshold value used to filter the results, etc.). Such data could be provided in supplemental data. In the description of the methodology of proteomic measurements, there is no description of how data from measurements were processed and analyzed (software, databases, setup of search parameters etc.)
Response #1: We added new Table S1, Table S2, and Table S3 for the data with Mascot score and peptide match. We used Mascot software and SwissProt database.
We slightly modified the text as follows: “against SwissProt database with the setting of 1 mis-cleavage and the fixed modification: Oxidation (Methionine), Carbamidomethyl (Cysteine).” (line 347-349), and “Histone H1.2 and GNL3 were identified with high reliable values (Mascot score, peptide match number) by Mascot software, “92, 8” and “36, 4” respectively.” (line 189-191)
Point #2: Table 2 does not contain accurate information about the identified proteins.
- Most of the symbols used are symbols used to describe genes, e.g. RPS6, GNL3. The results of proteomic research refers to amino acid sequences (products of the gene) and names of proteins must be used.
Response #2a: We corrected the name of all molecules in Table 2 to protein name, and added the UniProt accession numbers.
- some of the descriptions are very imprecise, e.g. there are several genes of the HSP70 protein in humans (Cell Stress & Chaperones. 1 (1): 23–8. doi:10.1379/1466-), which protein or proteins the authors identified?
Response #2b: HPS70 we identified in this experiment might be HSP70-1a (HSPA1A) or HPS70-1b (HSPA1B). It was impossible to discriminate these two molecules by mass spectrometry or other biochemical methods but protein sequencing. We referred this protein as Heat shock 70 kDa protein 1A/1B (HSP70-1A/B) (HSPA1A/B) in this manuscript.
- there are at least several hundred receptor proteins containing seven transmebrane helixes in the human body? What proteins from this family have been identified?
Response #2c: A seven transmembrane helix receptor protein we identified in this experiment was Pituitary adenylate cyclase-activating polypeptide type I receptor (P41586).We corrected the Table 2 and the text.
Summing up, the data contained in Table 2 are to be precise by giving a reference to the protein sequence (and not the names of genes), it is not the genes that are being studied in this case.
Modify the data in Table 2 (unify the naming and use of protein sequence names/symbols). It must cause changes to the text and descriptions of the drawings so that uniform naming is applied throughout the text.
Response: We described all the molecules by protein name. We also added the UniProt accession numbers.
Point #3: 10 out of 43 (references 1-2, 8-14, 43) literature items are self-citation. It can be assumed that the topic presented in the manuscript is very niche.
Response #3: As the reviewer pointed out, we cited several our paper in this manuscript. However, in References 1 and 2, we described the general remarks on oral tumors, and in Reference 43, we cited the results of STR analysis of the cells used. Therefore, we cited 7 papers concerning TSC-22 from our previous publications (ref:8-14). Because TSC-22 is a molecule that shows important role in development, proliferation, carcinogenesis, differentiation, apoptosis, and cellular senescence, many researchers all over the world have analyzed the function of TSC-22. At present, it has been clarified that four different genes produced several TSC-22 proteins, and these proteins forms super family. Among these 4 genes, we cloned human TSC-22 (TSC22D1-2) cDNA which was induced by an anticancer drug having differentiation-inducing activity. We clarified its structure and function, especially on the proliferation, differentiation, and cell death in salivary gland cancer cells. In this experiment, we expanded the research based on our previous results, then we need to cite our previous publications. In the revised manuscript, we cited several articles from many other laboratories published so far (Ref: 16, 18-20, 24, 28-37) in order to show that the study of TSC-22 was not a niche work.
The first part of Introduction lines 41-50 is completely redundant to the readability of the manuscript and should be removed.
Response: This part might be redundant for biochemists or molecular biologists. However, we think that this part is an essential to explain the specific aim of this study, in which we attempt to establish the differentiation-inducing therapy to oral cancer based on the current status of molecular-targeted therapy and differentiation-inducing therapy.
Lines 77-78 Among the TSC-22 family, the function of TSC-22 (TSC22D1-2) has been widely studied” – here should be references to this extensive research ?
Response: Yes, we added the several publications concerning TSC22D1-2 from our research group as well as other research groups (Ref: 16, 18-20, 24, 28-37)
Table 1. For most transcripts, the only reference is to the NCBI. However, all these transcripts have been described in scientific publications and it is these publications that should be cited as literature references. For example, the transcript of TSC22D1-4 (NP_001230728.1) is described in the publication Genet Med. 2017 Oct;19(10):1105-1117. doi: 10.1038/gim.2017.37. Epub 2017 May 11. In the description of the table, it must be stated that the accession numbers refers to the NCBI database (reference 17).
Response: We added the references and slightly modified Table 1. We cannot find any information concerning the transcript of TSC22D1-4 (NP_001230728.1) in the suggested publication.
Where in Figure 7 should be the band for G protein of the nucleolar 3 protein (GLN3 gene) ?
Response: As we described in the text (lines 170-171), GNL3 could not be detectable in the nuclear fraction and whole cell lysate in this experimental condition (results were not shown). We slightly modified the text, the appearance of “Fig. 7” and “data not shown”. However, GNL3 was detected in the in vitro TSC-22 pull-down sample (supplemental data). These results showed that TSC-22 had a potential to bind GNL3. We discussed the discrepancy of the results from in vitro data and in vivo data.
Reviewer 2 Report
This article by Kamimura et al. studied the cellular location of TSC22D1 family proteins and tried to identify their binding partners by mass spectrometry. The immunofluorescence results clearly show the location of TSC22D1-2 and TSC22D1-3, and their transition upon DNA damage. Several binding proteins were identified, within which, H1 and GNL3 were verified and considered as important partners in the cellular signal model speculated by the authors. After throughout evaluation of the research article, I personally felt that the presented article is good, and it fits with the scope of the IJBIOMAC due to the research work address an interesting topic for researchers. Some comments and suggestions are presented below to improve quality and to clarify some information.
- It is hard for reader to follow when the author used various cell lines to conduct different experiments. It will be nice if the authors could provide data with more consistent cell lines usage and provide the reason why the particular cell lines were chosen.
- I didn’t understand why the authors design the truncation of TSC-22N. If they were trying to claim TSC-22N will stay in cytoplasm upon DNA damage, please provide the data.
- Figure 5 is messed up with an extra A panel, please correct it .
Author Response
Response to Reviewer 2 Comments
Point 1: It is hard for reader to follow when the author used various cell lines to conduct different experiments. It will be nice if the authors could provide data with more consistent cell lines usage and provide the reason why the particular cell lines were chosen.
Response 1: In this study, we basically used two types of cells, TYS cells and HEK293 cells. Because we cloned TSC-22 cDNA from TYS cells, which is a salivary gland cell line, we used TYS cells for functional analysis. However, on the artificial overexpression system by transfection, there were several problems in TYS cells such as efficiency of gene transfection, influence of the endogenous gene on the integrated site, and expression efficiency of the transfected gene. Therefore, we used HEK293 cells and the Flp-InTRex system. Furthermore, we show the similar results of the intracellular localization of endogenous TSC-22 in the different cells revealed in this study on "Discussion" and "supplemental data”.
Point 2: I didn’t understand why the authors design the truncation of TSC-22N. If they were trying to claim TSC-22N will stay in cytoplasm upon DNA damage, please provide the data.
Response 2: We did not examine the alteration of the localization of TSC-22N after DNA damage. Because TSC-22N was constructed as a truncation mutant with NES but without LZ and TSC-box. We observed the function of NES sequence on the protein by its localization. TSC-22N was an artificial protein which did not exist naturally. Therefore, we did not used TSC-22N for subsequent experiment. On the other hand, because TSC-22 (86) exists naturally as described in the text, its specific function has been examined by subsequent experiments.
Point 3: Figure 5 is messed up with an extra A panel, please correct it.
Response 3: We corrected.
Round 2
Reviewer 1 Report
Thanks to authors, to responce to my comments. I'm satisfied with manuscript changes and authors replay to my comments.
Reviewer 2 Report
I am satisfied with the authors' reply.